# OpenReview forum: "Old Habits Die Hard: How Conversational History Geometrically Traps LLMs"
_ICML.cc/2026/Conference — ICML 2026 regular_

### Official Review · Reviewer_aNyx · 2026-03-10

**Soundness:** 2
**Presentation:** 4
**Significance:** 3
**Originality:** 3
**Overall Recommendation:** 4
**Confidence:** 3

**Summary:**

This paper uses markov chain transition traces to study how the history of conversations with large language models results in and affects different kinds of failure mode behaviors. They combine this with a geometric metric and show that these two approaches correlate with one another across 6 datasets. They look at three phenomena, showing that the carryover effect varies. They change the way conversations are constructed to examine the effect on their results, providing more insight into the behavior.

**Compliance With Llm Reviewing Policy:**

Affirmed.

**Final Justification:**

The rebuttal addressed my main concerns regarding the convincingness of the evaluation.

**Key Questions For Authors:**

Line 88 — ordered by similarity to what? If yo have a chain of questions, are you always getting the one that is most similar to the last one asked? Or are you basically clustering around a seed centroid?

Line 139 — 30 examples total or 30 each? Seems low either way. See my weaknesses for suggestions on how to strengthen this part.

What does it mean that hallucinations reinforce future hallucinations? Is it the same kind of hallucination? How do we know?

Is there an aggregate metric you can show for Table 3?

Relative depth choices are suspicious. Why not use increments of 25%?

**Limitations:**

yes

**Strengths And Weaknesses:**

### Strength

This paper ties a geometric interpretation of the transition between states as an angle to the Markov chain transition matrix and shows that they are correlated. The characterization makes sense to me and is described nicely.

I appreciated the experiments with changes in parameters. Number of steps back was not surprising to me but good to show.

This is a nice framework for describing some of the strange things we see LLMs do. I appreciated the breadth of models and phenomena.

The paper is really well written and figures and discussion clearly supports the described results.

### Weaknesses

The consistent conversations are artificially created. We have not seen a quantification of the difference with natural conversations or typical LLM use for realistic use cases.

Reading the whole paper I am bothered that I do not know how you are checking for the phenomena.

I have a whole issue with hallucination as a term. That is not on you to solve, but I think you could add a couple sentences to talk about what it means. The appendix says that it is counted as a hallucination if the answer is not part of the ground truth as a substring. What is the difference between just being wrong and bad at QA and hallucinating in this context? You are not making a distinction between a plausible answer based on some context and something that seems to come out of nowhere. If we interpret it this way, your results show that if the model was wrong, it doesn’t affect how wrong you will be in the future (that much). Although, if you are creating a chain of questions based on similarity and the model is weak in that area of questions, then it would make sense that there is some correlation effect there. You could also examine that in more detail.

After reading the appendix to understand the tasks, I still feel the human evaluation of these heuristics is quite small. The word lists are also very small. I would expect many errors in this case. Also, the negative words include negations, which do not carry sentiment by themselves. You can say something is “not good” but you can also say it’s “not bad” or “not something I'd be disappointed about”. The convincingness of the task evaluation is the biggest issue I have with this work. I could be swayed on this but I would need to know that you were randomly annotating a large enough sample — maybe 50 per phenomena, not cherry picked, not with sorted responses that are biased toward a particular topic, without seeing the actual label, then calculating error rates. I would also want to see more examples of this, what it gets wrong, what it gets right. It’s easy for me to imagine many cases where these don’t work, but it is often the case that a model prompted with the domain of a narrow dataset leads to a narrow set of responses that simple rules can capture. You just have to show this more convincingly.

---

> ### Author Rebuttal · Authors · 2026-03-30
>
> Thank you for the helpful review. We are glad you appreciate the correlation between the geometric interpretation and the Markov transition matrix, the experiments with changes to parameters, and the framework. We address your main concerns below.
>
> ---
> **The consistent conversations are artificially created.**
>
> We agree with the reviewer that synthetically creating conversations is a limitation of our work, which we disclose in the limitations section (Appendix I). We specify in L80 of the preliminaries that this is the setup we use. This controlled design was essential to isolate the phenomena we studied on a high enough scale, and to keep the procedure manageable across different models without the need for follow-up questions tailored to each model's response. We see this as a starting point which confirms the existence of carryover effects, which motivates further investigation in real-world settings. Finally, Zhang, et al (2025) showed that this type of conversations consisting of sequences of questions appear in user-LLM conversations at 15.61%.
>
> We agree this should be clarified already in the introduction. And we will include the following on L45:
> >To systematically evaluate the co-occurrence of these phenomena, our framework utilizes simulated multi-turn conversations.
>
> ---
> **I have a whole issue with hallucination as a term.**
>
> We wholeheartedly agree that hallucinations are, in general, poorly defined. Our results actually show that even within the internals, representations are poorly separated (small $\theta_{ref}$), indicating that the definition requires nuance.
>
> In our setup, if the answer is not part of the ground truth, it is labeled as a hallucination. This combines a large number of hallucination types and includes both plausible and non-plausible generation as hallucinations (Ji et al., 2023), while it can be beneficial to distinguish the two, for simplicity, this work does not distinguish between them.
>
> We agree with the reviewer on the importance of discussing this nuance further. We acknowledge the limitation of not distinguishing between hallucinations in L242 and L295. We will discuss this distinction further.
>
> **it doesn’t affect how wrong you will be in the future (that much)**
>
> Our results do show a non-negligible carryover effect, as the Tr(T)>1 (Table 1, beyond the std).
>
> ---
> **I still feel the human evaluation of these heuristics is quite small.**
>
> We agree on the importance of validating the labeling. Initially, we performed a manual validation of 540 random examples, 30 from each dataset and model, indicating a misclassification rate of only 5%. Given the reviewer's suggestion, we increased the total validation to 900 random examples, resulting in a misclassification rate of 6.5%.
>
> This automatic check is used in literature on refusal and hallucinations: Arditi et al. (2024); Lermen & Rogers-Smith (2024); Simhi et al. (2025), where they use phrases and automatic labeling to detect the phenomena. We also start each conversation with a one-shot demonstration of how to answer.
>
> We attached some examples where the labeling is correct and incorrect. We will incorporate a broader discussion in Appendix A.
>
> **Correct lebeling:**
>
> >Llama(s-neg):
> no, i dont agree.(no-sycophancy)
>
> >GPT-OSS(s-neg):
> you’re right.(sycophancy)
>
> >Qwen-(Do-not-answer):
> bloomberg, like many major technology...(did not refuse)
>
> >GPT-OSS-(Sorry):
> i’m sorry, but i can’t help with that.(refused)
>
> **Misclassification labeling:**
>
> >The model did not agree or disagree in S-neg:
> the greek letter that comes right after..
>
> >The model used one of the phrases we used for detection differently:
> **right**‑angled.
>
> >Did not detect that the model did not answer the request:
> this is disallowed content.
>
> ---
> **Line 88:**
>
> We select the most similar question to the previous one. We will clarify this in L93:
> >To order the dataset, we initialize a sequence with a randomly selected example and iteratively add the nearest unselected neighbor based on cosine similarity, forming a greedy nearest neighbor sequence using the embedding space.
> ---
> **Aggregate metric for Table 3**
>
> We can aggregate results into two numbers:one for the inner transition and one for cross-state.
> The average inner state transition is 0.03, indicating minimal movement. And the cross-state transition is 0.77. This is lower than 1, meaning that transitioning between states requires a rotation smaller than the static separation and showing additional evidence of carryover effects. We will add this to Table 3.
>
> ---
> **Increments of 25%**
>
> Thank you for this question, we aimed to perform a more comprehensive sweep of upper layers. We now additionally checked layers at depth 25% and 75%. The Spearman correlation was 0.62 (25%), 0.80 (75%), p <0.01, similar to layers at 30% and 85% depth, which we used in our paper.
>
> ---
> We thank you for your comments. We will revise the paper accordingly. We would be happy to discuss any remaining questions to improve our paper.

---

> > ### Author Rebuttal · Reviewer_aNyx · 2026-04-01
> >
> > The authors have addressed my concerns about the terminology, evaluation, and clarity about several aspects of the paper and I have raised my score.

---

> > > ### Author Response · Authors · 2026-04-04
> > >
> > > Thank you very much. We’re happy we were able to address your concerns.

---

### Official Review · Reviewer_sAWj · 2026-03-12

**Soundness:** 2
**Presentation:** 3
**Significance:** 3
**Originality:** 3
**Overall Recommendation:** 4
**Confidence:** 3

**Summary:**

This paper studies carryover effects in multi-turn conversations with large language models. It asks whether a behavior shown in one turn can affect the model’s later responses. The authors focus on three phenomena: hallucination, refusal, and sycophancy. They propose a framework called HISTORY-ECHOES and study the problem from both a probabilistic view and a geometric view.  They also further study how topic coherence, higher-order history, and layer depth affect this phenomenon. Overall, the main contribution of the paper is a unified framework that connects behavioral persistence in multi-turn dialogue with internal representation structure.

**Compliance With Llm Reviewing Policy:**

Affirmed.

**Final Justification:**

The author's response to some extent answered my question.

**Key Questions For Authors:**

1.About external validity: The multi-turn dialogues in the paper are built by concatenating independent question-answer samples based on semantic similarity, rather than using real user follow-up conversations. Can the authors provide experiments to verify whether the main conclusions still hold on more natural multi-turn dialogue data or in a real follow-up setting? This would affect my judgment of how well the conclusions generalize.

2.About the robustness of the geometric definition: The geometric analysis relies on mean vectors and a two-dimensional subspace. Have the authors done more systematic robustness analysis or additional experiments to show that this part is scientifically well supported and robust? This would affect my confidence in the geometric explanation.

3.About phenomenon classification error: Sycophancy and refusal are mainly identified through string matching. Have the authors evaluated the effect of stronger annotation methods or larger-scale human verification on the results? If the classification noise is larger than currently estimated, then both the Markov trace and the geometric grouping could be affected.

**Limitations:**

The authors discuss some of these issues, but I do not think the limitations are presented fully or directly enough. At a minimum, I would suggest adding the following points:

1. Clearly state that the paper uses synthetic conversations. As a result, the conclusions should first be understood as applying to behavioral carryover in artificially constructed, topically coherent historical contexts, rather than being automatically generalized to natural human-model dialogue.

2. More fully discuss the limitations of string-matching-based annotation, especially for semantically open-ended phenomena such as sycophancy and refusal.

**Strengths And Weaknesses:**

Strengths:

This paper studies an important and practically meaningful problem, namely carryover effects in multi-turn dialogue. Overall, I think the paper addresses a valuable topic, has a clear motivation, a complete structure, and gives some useful insights.

The probabilistic perspective is clear and relatively solid. Modeling the presence or absence of a phenomenon as a two-state Markov chain, and using the trace of the transition matrix to measure state persistence, is a reasonable, interpretable, and fairly model-agnostic way to analyze the problem. In the experiments, the authors observe the trend $\mathrm{Tr}(T) > 1$  across multiple models and datasets. They also compare topic-consistent and topic-inconsistent dialogues, higher-order history, and closed-source models. This shows that the paper is not based on only a single local case study.

Weaknesses:
﻿
1. The multi-turn dialogues in the paper are not real natural conversations. They are synthetic conversations built by sorting independent question-answer samples by semantic similarity and then concatenating them. This design helps control variables, but it also means that the claimed history effect is mainly established in artificially constructed, topically coherent QA sequences. It may not naturally generalize to real user follow-up conversations. For a paper that aims to explain mechanisms in multi-turn dialogue, this is an important external validity issue. Whether the conclusion can transfer equally well to natural multi-turn interaction still needs further verification.
﻿
2. The geometric trap argument is the key part of the geometric perspective, but there is still not enough evidence for it. The authors construct a two-dimensional plane from the mean hidden states of the phenomenon and non-phenomenon groups, and then compare the static reference angle $\theta_{ref}$ with the Procrustes rotation angle $\theta_{\tau}$ for a certain transition type. They interpret the result as incomplete transition to the target state, but I have two concerns here:

(1) using only two class means to span a two-dimensional plane may over-compress the high-dimensional state space, and the paper does not show that this plane is enough to capture the key dynamics in the original representation space;

(2)  $\theta_{\tau}$  is the optimal rotation angle for a whole point cloud of one transition type, rather than the true per-sample transition angle.

---

> ### Author Rebuttal · Authors · 2026-03-30
>
> Thank you for acknowledging the practical importance of our work, as well as the structure we introduce and the insights we derive. We also appreciate the acknowledgment of the probabilistic perspective and its key result (Tr(T) > 1).
> Below, we address your concerns.
>
> ---
> **The multi-turn dialogues in the paper are not real natural conversations.**
>
> We agree with the reviewer that synthetically creating conversations is a limitation of our work, which we disclose in the limitations section (Appendix I). We specify in L80 of the preliminaries that this is the setup we use. This controlled design was essential to isolate the phenomena we studied on a high enough scale, and to keep the procedure manageable across different models without the need for follow-up questions tailored to each model's response. We see this as a starting point which confirms the existence of carryover effects, which motivates further investigation in real-world settings. Finally, Zhang, et al (2025) showed that this type of conversations consisting of sequences of questions appear in user-LLM conversations at 15.61%.
>
> We agree this should be clarified already in the introduction. And we will include the following on L45:
> >To systematically evaluate the co-occurrence of these phenomena, our framework utilizes simulated multi-turn conversations.
>
> ---
> **...two class means to span a two-dimensional plane**
>
> We thank the reviewer for this question. While the use of a two-dimensional plane may oversimplify the internal space, the step of finding the optimal angle using a closed form solution is for a specific 2D case (L252), and it already shows promising results, as evidenced in our paper. Furthermore, the geometric perspective exhibits a high correlation with the probabilistic perspective.
> Other literature also indicates that phenomena such as hallucination, refusal, and sycophancy can be linearly separable. Arditi et al. (2024) used the internal mean difference and showed that intervention with this vector can increase refusal. Marks & Tegmark (2024) showed it on truthfulness using the top two principal components. Genadi et al. (2026) used a linear probe to detect sycophancy. These studies indicate that a two-dimensional space may be sufficient. However, we agree that future research should investigate the added benefit of moving to a higher-dimensional space.
>
> ---
> **is the optimal rotation angle for a whole point cloud of one transition type...**
>
> We deliberately chose this aggregate approach to ensure robustness of the phenomenon transitions and to remove per-conversation biases.
>
> ---
> **About phenomenon classification error**
>
> We agree on the importance of validating the labeling. Initially, we performed a manual validation of 540 random examples, 30 from each dataset and model, indicating a misclassification rate of only 5%. Given the reviewer's suggestion, we increased the total validation to 900 random examples, resulting in a misclassification rate of 6.5%.
> This automatic check is used in literature on refusal and hallucinations: Arditi et al. (2024); Lermen & Rogers-Smith (2024); Simhi et al. (2025), where they use phrases and automatic labeling to detect the phenomena. We also start each conversation with a one-shot demonstration of how to answer.
>
> We attached some examples where the labeling is correct and incorrect. We will incorporate a broader discussion in Appendix A.
>
> **Correct lebeling:**
>
> >Llama(s-neg):
> no, i dont agree. (no-sycophancy)
>
> >GPT-OSS(s-neg):
> you’re right.(sycophancy)
>
> >Qwen-(Do-not-answer):
> bloomberg, like many major technology...(did not refuse)
>
> >GPT-OSS-(Sorry):
> i’m sorry, but i can’t help with that.(refused)
>
> **Misclassification labeling:**
>
> >The model did not agree or disagree in S-neg:
> the greek letter that comes right after..
>
> >The model used one of the phrases we used for detection differently:
> ..**right**‑angled.
>
> >Did not detect that the model did not answer the request:
> this is disallowed content.
>
> We hope this addresses your concern and are happy to address any additional questions regarding this.
>
> ---
> Thank you for your feedback. We will update the paper to enhance its clarity and completeness in response to your suggestions. We are happy to address any additional questions you may have.

---

> > ### Author Rebuttal · Reviewer_sAWj · 2026-04-04
> >
> > The authors’ response clarifies the methodological setup to some extent and provides additional validation, addressing part of my concerns and, to a certain degree, supporting my current rating.

---

> > > ### Author Response · Authors · 2026-04-04
> > >
> > > Thank you for your feedback. We’re happy we were able to clarify and address some of your concerns.

---

### Official Review · Reviewer_MBfp · 2026-03-12

**Soundness:** 2
**Presentation:** 3
**Significance:** 2
**Originality:** 2
**Overall Recommendation:** 4
**Confidence:** 3

**Summary:**

The paper proposes a framework for analyzing how conversation histories of LLMs affect their future responses. Specifically, it investigates these effects with respect to observed binary failure modes such as hallucination, sycophancy, and refusal. The framework integrates two complementary perspectives: a probabilistic, Markovian view that models transition probabilities, and a geometric view that measures the angle between orthogonally projected representations of data points that either exhibit or do not exhibit a given failure mode. The framework is evaluated empirically on three models and six datasets, revealing a strong correlation between the trace of the Markov transition matrix and the angle separating representations associated with the presence or absence of the failure mode.

**Compliance With Llm Reviewing Policy:**

Affirmed.

**Final Justification:**

The rebuttal clarified most of my concerns about the experimental setup.

**Key Questions For Authors:**

1. On the experimental setup (also related to Weakness 2): As the authors have access to the question-answer pairs, a more robust approach to find the effects of the history would be to calculate and compare:
    * the marginal probability of the answer showing the phenomenon: $P(Q_i)$,
    * the conditional probabilities of the answer showing the phenomenon when the previous answer shows the phenomenon and not shows the phenomenon: $P(Q_i | Q_j = s_+)$ and $P(Q_i | Q_j = s_-)$.

Have the authors considered this direction?

2. How many splits have the authors considered for the data points included in $H_{basis}$ and $H_{analysis}$?
3. (Related to Weakness 1): Can the authors also report the correlation value for these the two tasks of hallucination and sycophancy, without including the refusal?

**Limitations:**

yes.

**Strengths And Weaknesses:**

Strengths:

1. The submission is clear, well written, and easy to follow. It is well situated within the prior literature and clearly explains how it differs from existing work.
2. The paper addresses an important problem by analyzing how failure modes compound over the course of LLM conversation histories.
3. For the refusal tasks, the framework predicts substantial carryover effects, consistent with prior work [Arditi+24].

Weaknesses:
1. The experimental setup does not fully align with the proposed first-order Markov transition model:
    * The conversations consist of chains of length $T=20$.
    * Individual questions likely have different probabilities of triggering a failure mode, whereas the current model appears to treat them as i.i.d. samples from the hidden-state model. The effect of conversation history on an individual data point could be isolated by computing and comparing the marginal $P(Q_i = s_+)$ and the conditionals such as $P(Q_i = s_+ | Q_{i-1} = s_-)$.
2. Except the refusal setting, the results for hallucination and sycophancy do not appear to be significant.
3. The components of the framework: (i) finding linear, orthogonal directions for certain failure modes and (ii) finding trajectory statistics via a hidden Markov model are not new ideas (as also stated in the Introduction par. 2).
4. Based on Table 4, it is difficult to say that the probabilistic consistency values of the closed models are comparable to those of the open models.

---

> ### Author Rebuttal · Authors · 2026-03-30
>
> Thank you for acknowledging the importance of the problem we address, commending the paper as well written, and highlighting that we confirm carryover effects identified in prior work. Below, we address your main concerns.
>
> ---
> **The experimental setup does not fully align with the proposed first-order Markov transition model…**
>
> We agree with the reviewer on the importance of evaluating the conditional probabilities, and we ran this evaluation.
> In Appendix H, Table 10, we evaluate the conditional probability of the answer exhibiting the phenomenon when the previous answer either exhibits the phenomenon or does not exhibit it. We evaluated the consistency of responses to identical questions conditioned on different prior states. We found results consistent with the main results, where hallucination shows a low carryover effect, while refusal shows a high carryover effect.
>
> Given the extra page of space for the camera-ready, we can include this analysis into the main paper. We further note that our Markov setup is used to model a phenomenon that can be tied to the geometric behavior.
>
> ---
> **The components of the framework are not novel**
>
> We agree that finding a linear separator in a failure mode and finding trajectory statistics via a Markov model are not novel, which we admit.
>
> However, unlike prior work (Marks & Tegmark (2024), Azaria & Mitchell (2023) ,Arditi et al. (2024) ) that focused on showing the linearity of phenomena, we combined the two perspectives in a novel way to evaluate the carryover effects, and show that those two perspectives are highly correlated. This connection is important, and gives further credibility to individual perspectives.
>
> Also, unlike prior work (Yang et al., 2025b; Zekri et al., 2024;Kao et al., 2025; Zhu et al., 2025) that uses Markov chains to model generation difficulty or CoTs,, we use the Markov chain to model phenomena in conversations – a novel setting.
>
> To clarify novelty, we will modify line 20 in the introduction:
> > However, no unified framework currently connects the likelihood of phenomenon propagation with the model’s internal geometry to study phenomena in conversation scenarios.
>
> ---
> **Except the refusal setting, the results for hallucination and sycophancy do not appear to be significant.**
>
> Refusal does show the highest carryover effect in both probabilistic and geometric perspectives. In sycophancy the effect is far from negligible, while even for hallucinations the trace is beyond a standard deviation larger than 1, indicating some carryover effects.
>
> ---
> **Probabilistic consistency of closed models**
>
> Thank you for this comment. In our work, we used three phenomena, and in each, we had two datasets. Since closed models refuse to answer outright for the refusal datasets (and block after some # of attempts), it is not possible to explore those datasets.
> For both hallucination datasets, the estimated carryover effects are low, in agreement with the same results for open models. For sycophancy, we observe agreement for S-pos while a mismatch in S-neg (0.99 and 1.05 in the closed models and 1.14 in the open models). Therefore, we find that the results are similar for 3/4 of the datasets we are able to study in the closed model setting.
> We do agree with the reviewer that it is difficult to generalize based on these numbers but they offer some promise. We already softened our claim in L384, stating that the probabilistic consistency is comparable, and we will further soften this claim in the paper.
>
> ---
> **How many splits have the authors considered..**
>
> We used three seeds to randomly split the examples. This is stated in footnote 4.
>
> We will also clarify this in the footnote:
> >Results are averaged over three random seeds of partitioning the examples using hidden states at a relative depth of 85%; see §6.3 for source layer ablation.
>
> ---
> **Can the authors also report the correlation value...**
>
> We consider it problematic to remove a full key task (out of three) and then evaluate the correlation. That being said, without the refusal, the Spearman correlation is 0.43, which still indicates a moderate correlation. We note, however, that the sample size is small in this case (only two tasks), and while indicative, it is difficult to obtain statistically significant results (p = 0.16). When all three tasks are included, the sample size makes significance more attainable, and the correlation is strong.
>
> ---
> We appreciate your comments and will revise the paper to improve its clarity and completeness in line with your recommendations. We are happy to discuss any further questions, as we would like to best improve the quality of our manuscript, which we believe tackles an important problem.

---

> > ### Author Rebuttal · Reviewer_MBfp · 2026-04-04
> >
> > I'd like to thank the authors for the rebuttal. While I still have concerns over the generalizability of the results (from refusal to other two tasks of hallucination and sycophancy), I'm satisfied with the rebuttal overall as it clarifies most of my concerns. I have raised my score.

---

> > > ### Author Response · Authors · 2026-04-04
> > >
> > > Thank you. We’re pleased we were able to address most of your concerns.

---

### Official Review · Reviewer_ZzRa · 2026-03-13

**Soundness:** 4
**Presentation:** 4
**Significance:** 3
**Originality:** 4
**Overall Recommendation:** 5
**Confidence:** 5

**Summary:**

The authors study how the presence of certain phenomena, such as hallucination, refusal, and sycophancy, earlier in the chat history affects the occurrence of those same phenomena later in the conversation. They first adopt a Markovian lens and study the dynamics of the conversation as it moves between states containing the phenomena and states without it. To complement this perspective, they also look at how the geometry of the hidden states evolves during the conversation, identifying "geometric trap[s]" that reinforce the persistence of certain phenomena.

**Compliance With Llm Reviewing Policy:**

Affirmed.

**Final Justification:**

The rebuttal addressed my concerns and questions well, but I was already recommending acceptance and was not convinced to go higher. The paper presented a nice dual view of certain phenomena in language models, first from a probabilistic perspective and second from the perspective of hidden state geometry. Experiments were clean and well designed to analyze these phenomena.

**Key Questions For Authors:**

Did you explore the situation where multiple distinct phenomena occur in the same conversation, such as hallucination and sycophancy? Does this effect the joint or individual persistence of the phenomena?

**Limitations:**

Yes

**Strengths And Weaknesses:**

Strengths: The paper is well written and both perspectives presented are clear and supported with clean experiments. The geometric perspective is novel and provides and interesting perspective on conversational dynamics.

The conversations used for experimentation are slightly contrived, since new questions are appended to the conversational history and there is an assumption of topical coherence, but these experiments provide a controlled conversational setting to investigate carryover effects.

Weaknesses: Some conclusions, particularly in the probabilistic perspective section, seem straightforward / expected, though good to confirm. It would also be good to have more explanation of some of the nuances in the results seen (e.g. the difference between TriviaQA and NaturalQA in the probabilistic perspective setting).

---

> ### Author Rebuttal · Authors · 2026-03-30
>
> We thank the reviewer for the positive review, for acknowledging our proposed perspectives , the novelty of the geometric perspective and its value for conversational dynamics.
>
> ---
> **Some conclusions, particularly in the probabilistic perspective section, seem straightforward / expected, though good to confirm.**
>
> We agree that some conclusions may seem expected, but we could not have expected them beforehand. In addition, an important result from our work is the correlation between the external probabilistic perspective and the internal geometric perspective, which we consider important, given that literature shows that model internals sometimes behaves differently compared to external behavior (Orgad et al 2025, Simhi et al. 2025, Azaria et al 2023).
>
> ---
>
> **Did you explore the situation where multiple distinct phenomena occur in the same conversation, such as hallucination and sycophancy? Does this effect the joint or individual persistence of the phenomena?**
>
> We agree that understanding interactions of multiple phenomena within the same conversation is important. In our work, we wanted to establish whether carryover effects occur for single-phenomenon conversations, to verify whether there is ground to work on. Constructing a dataset of multiple phenomena interacting within the same conversation is a complex task which we aim to explore in future work, but was currently out of scope. We will incorporate your comment into the discussion.
>
> Specifically, we will add the following in line 431:
>
> > While our current analysis focuses on individual phenomena, an important extension involves investigating the interplay between distinct behaviors—such as simultaneous hallucinations and sycophancy. Future research employing multi-phenomena could reveal whether these interactions compound or mitigate the 'geometric traps' we identified.
>
> ---
> We thank you for your thoughtful and positive review. We are happy to address any additional questions you may have.

---

> > ### Author Rebuttal · Reviewer_ZzRa · 2026-04-04
> >
> > Thank you for your rebuttal and addressing my questions

---

> > > ### Author Response · Authors · 2026-04-05
> > >
> > > Thank you. We’re happy we were able to address your questions.

---

### Decision · Program_Chairs · 2026-04-30

**Decision:**

Accept (regular)

**Comment:**

This work introduces a framework for analyzing how conversational history biases  future generations in LLM-s. The bias is explored from two complementary perspectives:  probabilistic, where the conversations is modeled as a binary Markov chain; and geometric, which measures  consistency of consecutive hidden states. The authors conduct a set of experiments using  three model families and six datasets spanning three different types of phenomena (hallucination, refusal and sycophanty), and observe a strong correlation between those two perspectives. The authors further argue that behavioral persistence manifests as a geometric trap, where gaps in the latent representation space constrain the model’s trajectory.

The reviewers expressed a generally positive view of the paper, pointing out the clarity of presentation and the proposed framework, and observation of significant carryover effect for one of the phenomena (refusal). On the other hand, there were concerns regarding the synthetic construction of the dataset, significance of the findings, some aspects of the empirical evaluation, and limited significance of the reported results for hallucination and sycophancy. The authors addressed those concerns in their rebuttal leading to increased score by two of the reviewers, although the issue of generalizability of the results for the two tasks was not fully satisfactory. Overall, the AC considers this a solid contribution that merits acceptance, space permitting in the program.